# Mobile Laboratory Reveals the Circulation of Dengue Virus Serotype I of Asian Origin in Medina Gounass (Guediawaye), Senegal

**DOI:** 10.3390/diagnostics10060408

**Published:** 2020-06-16

**Authors:** Idrissa Dieng, Boris Gildas Hedible, Moussa Moïse Diagne, Ahmed Abd El Wahed, Cheikh Tidiane Diagne, Cheikh Fall, Vicent Richard, Muriel Vray, Manfred Weidmann, Ousmane Faye, Amadou Alpha Sall, Oumar Faye

**Affiliations:** 1Arboviruses and Hemorrhagic Fever Viruses Unit, Virology Department, Institut Pasteur de Dakar, BP220 Dakar, Senegal; idrissa.dieng@pasteur.sn (I.D.); MoussaMoise.Diagne@pasteur.sn (M.M.D.); ct.Diagne@pasteur.sn (C.T.D.); Cheikh.Fall@pasteur.sn (C.F.); Amadou.SALL@pasteur.sn (A.A.S.); 2Epidemiology Unit, Institut Pasteur de Dakar, BP220 Dakar, Senegal; ghedible@pasteur.sn (B.G.H.); vrichard@pasteur.nc (V.R.); muriel.vray@pasteur.sn (M.V.); 3Microbiology and Animal Hygiene, University of Goettingen, D-33077 Goettingen, Germany; abdelwahed@dpz.eu; 4Institute of Aquaculture, University of Stirling, Scotland FK9 4LA, UK; m.w.weidmann@stir.ac.uk

**Keywords:** fever, NMFI, mobile laboratory, RPA, DENV

## Abstract

With the growing success of controlling malaria in Sub-Saharan Africa, the incidence of fever due to malaria is in decline, whereas the proportion of patients with non-malaria febrile illness (NMFI) is increasing. Clinical diagnosis of NMFI is hampered by unspecific symptoms, but early diagnosis is a key factor for both better patient care and disease control. The aim of this study was to determine the arboviral aetiologies of NMFI in low resource settings, using a mobile laboratory based on recombinase polymerase amplification (RPA) assays. The panel of tests for this study was expanded to five arboviruses: dengue virus (DENV), zika virus (ZIKV), yellow fever virus (YFV), chikungunya virus (CHIKV), and rift valley fever virus (RVFV). One hundred and four children aged between one month and 115 months were enrolled and screened. Three of the 104 blood samples of children <10 years presented at an outpatient clinic tested positive for DENV. The results were confirmed by RT-PCR, partial sequencing, and non-structural protein 1 (NS1) antigen capture by ELISA (Biorad, France). Phylogenetic analysis of the derived DENV-1 sequences clustered them with sequences of DENV-1 isolated from Guangzhou, China, in 2014. In conclusion, this mobile setup proved reliable for the rapid identification of the causative agent of NMFI, with results consistent with those obtained in the reference laboratory’s settings.

## 1. Introduction

In Africa, fever is the most common symptom leading patients to seek health care [1,2]. Fever of unknown origin has long served as an entry point for the treatment of malaria [3]. With encouraging gains in malaria control in Sub-Saharan African countries, the incidence of this disease is in decline, leading to a decreasing proportion of febrile illness attributable to malaria. Between 2000 and 2013, malaria mortality rates decreased by 47% globally, and by 54% in sub-Saharan Africa—the region most affected by the disease—whereas the proportion of patients presenting with non-malaria febrile illness (NMFI) increased, respectively [4]. Acute febrile episodes are caused by various bacterial and viral pathogens, and infections with these agents result in patients presenting with malaria-like symptoms [5]. Although resulting in a higher mortality than malaria, NMFIs are not being reliably diagnosed due to the lack of accurate, rapid and affordable diagnostic tests, and also due to poor access to diagnostics facilities in many resource-poor endemic settings [1,6,7]. The objective of this study was to determine potential arboviral etiologies of NMFI in children in a low resource setting, using mobile recombinase polymerase amplification (RPA)—a real-time isothermal amplification technique [8]. For this purpose, we conducted a prospective arbovirus investigation in children seeking healthcare, at a health centre in the Dakar suburb of Medina Gounass, during a period of six months—from September 2015 to March 2016. The mobile suitcase laboratory has also been successfully used for Ebola virus detection [9].

## 2. Materials and Methods

### 2.1. Study Site

For a pilot study, prospective molecular screening on NMFI was conducted between September 2015 and March 2016, at the “Institut de pédiatrie sociale”, located in the suburb of Dakar (Figure 1)—the capital city of Senegal, West Africa. Built in 1971, this health centre located at Pikine-Guediawaye has an outpatient department with care activities focused on mother and child health. It is also involved in the national program on immunization, in nutritional programs and in family planning. With 22 qualified staff, the health centre has 6 consultation rooms (including one for vaccination), a laboratory and a nutritional service. With around 1,000,000 inhabitants, Pikine-Guediawaye is an agglomeration of well-established traditional villages, and interspersed recent settlements, the latter mostly located in flood-prone areas, where housing is officially forbidden. The western part of these towns is located on the edge of a vast area of permanent marshland (Grande Niaye), where natural marshy hollows and furrows dug for market garden irrigation, as well as areas of prolonged stagnation of rainwater, are observed year-round. With a high density of housing (9200 inhabitants/km) in proximity to stretches of water and stagnant wetlands, the population lives in an under-serviced peripheral area, in crowded conditions, with poor water supply and sanitation, and dirt paths between dwellings and open sewers.

### 2.2. Patient Selection

Children less than 10 years old were enrolled if they met the following criteria: acute fever (≥37.5 °C axillary temperature), negative for malaria rapid diagnostic tests (RDTs) and living in the same area for four successive calendar months. Regarding the eligibility for enrolment, the study information was read to the legal guardian, and after obtaining informed consent, clinical symptoms were recorded, and 2 mL of venal blood was collected. The study initially consisted of a weekly visit every Monday, before Friday was added to compensate for the small sampling observed at the beginning of the survey.

### 2.3. Screening Procedure in the Field

Blood samples were collected and processed on site, using a mobile suitcase laboratory for viral identification. The mobile laboratory consisted of a glovebox (Bodo Koennecke, Berlin, Germany), a lab-in-a-suitcase [9,10,11], a solar panel and a power pack set (Yeti 400 set, GOALZERO, South Bluffdale, UT, USA, Figure 2). The mobile setup was organized into 2 stations: the extraction station with the glovebox, and the RPA in the suitcase laboratory. Inactivation and extraction of the samples were performed in the glovebox using a modified protocol of the Speedxtract kit (Qiagen, Lake Constance, Germany).

The extraction was performed by adding 100 µL of lysis buffer, 20 µL of sera and 30 µL of magnetic beads to a 1.5 mL tube, followed by incubation at 95 °C for 10 min. After incubation, the tube was transferred to a magnetic stand for 2 min, and after sedimentation of the beads, a 150 µL volume of supernatant was collected in a new 1.5 mL tube. RNA amplification of all samples was carried out using the Tubescanner point of care device (Qiagen, Hilden, Germany), and the Twist exo RT kit (TwistDx, Cambridge, UK), using RPA amplicons designed for the detection of 5 arboviruses: dengue virus (DENV), with two sets of primers and probes (set 1 for DENV1-3 and set 2 for DENV- 4 [10]); yellow fever virus (YFV) [12]; chikungunya virus (CHIKV) [13]; rift valley fever virus (RVFV) [14]; and zika virus (ZIKV) [15], with analytical sensitivities of 241, 14, 10, 23, 10, 21; and RNA molecules were detected, respectively. RT-RPA reactions were performed in a volume of 50 µL. Briefly, a mix containing 29.5 μL of rehydration buffer, 7.2 μL of ddH_2_O, 420 nM of each primer, and 210 nM of a target specific RPA exo-probe, was dispensed into each of the eight 1.5 mL tubes, before adding 5 μL of RNA template. Finally, 46.5 μL of master mix/template solution was transferred to each lyophilized RPA pellet of the eight-tubes strip provided in the kit. An amount of 3.5 μL magnesium acetate (280 mM) was added into the lid of each tube, before closing it and spinning the drop into the reaction mix. For real-time fluorescence monitoring, the reaction tubes were placed in the ESE Quant Tubescanner (Qiagen Lake Con- stance GmbH, Stockach, Germany).

### 2.4. Standard Assays in the Central Laboratory

All of the samples collected in the field were shipped to the WHO collaborating centre for arboviruses and haemorrhagic fever viruses, at the Institut Pasteur de Dakar (IPD), in order to perform complementary confirmatory tests.

#### 2.4.1. RNA Extraction and Real-Time qPCR

Viral RNA was extracted from 100 μL of human sera, using the QIAmp viral RNA kit (Qiagen, Hilden, Germany) according to the manufacturer’s instructions. The reduced input of 100 µL serum for this kit had been validated in-house. The RNA was eluted in 60 μL of elution buffer and placed on ice. To confirm the detection obtained in the field, real-time RT-qPCR assays for DENV [16], CHIKV [17], RVFV [18], YFV [19], ZIKV [20] were performed using the Quantitect Probe RT-PCR Master Mix (Qiagen). Briefly, the detection was performed using ABI7500, using the following temperature profiles for all RT-qPCR assays: RT at 50 °C for 10 min, activation at 95 °C for 15 min and 45 cycles of 2-step PCR— at 95 °C for 15 sec and 60 °C for 1 min.

#### 2.4.2. Non-Structural Protein 1 (NS1) Antigen Capture

The NS1 antigen assay was performed using the Platelia^TM^ Dengue NS1 Ag-ELISA kit (Biorad Laboratories, Marnes-La-Coquette, France), according to the manufacturer’s instructions. The capture and revelation steps were performed using a murine monoclonal antibody (MAb). The presence of the NS1 antigen in a sample was assessed by the formation of an immune-complex MAb-NS1-MAb (peroxidase). Briefly, 50 μL of the serum sample, 50 μL of the sample diluent (Diluent R7), and 100 μL of the diluted conjugate were added to each well precoated with the anti-NS1 monoclonal antibody. For each assay, positive and negative controls, as well as calibrator sera, were included. The plate was incubated for 90 min at 37 °C. After six wash steps—using 250 µL of washing solution (R2)—160 µL of tetramethylbenzidine (TMB) substrate was added to each of the wells, and the plate was further incubated at room temperature for 30 min in the dark, followed by the addition of 100 μL of stop solution (1N H2SO4). With the spectrophotometer optical density (OD) readings were measured at wavelengths of 450 nm/620 nm, were obtained and the index of each sample was calculated with the ratio OD of samples/OD of calibrators. Sample ratios <0.5, between 0.5 and 1.0, and >1.0 were classified as negative, equivocal, and positive, respectively.

#### 2.4.3. ELISA IgM Detection

We determined the presence of DENV, YFV, RVFV, CHIKV and ZIKV IgM in our samples by a capture enzyme-linked immunosorbent assay (MAC ELISA), following a published protocol [21]. For the coating step, a monoclonal capture antibody (anti human IgM) was added to a 96-well microtiter plate, and incubated at 4 °C overnight. The human sera were heat-inactivated (56 °C, 30 min), and screened at a dilution of 1:100 in phosphate-buffered saline (PBS), supplemented with 0.05% Tween and 1% milk powder. After washing the plate three times with PBS plus 0.05% Tween 20, a 1/100 dilution of the different serum samples and controls were added in duplicate to the plate and incubated at 37 °C for 1 h. The wells were washed three times, specific and non-specific antigens were deposited, and the plate was incubated for 1 h at 37 °C. After another washing step, the specific immune ascites were added to each well. After incubation and washing, a conjugate (peroxidase labelled antibody specific to mouse IgG) was added and allowed to react for 1 h at 37 °C. A tetra-methylbenzidine (TMB) substrate was added to the IgM conjugate complex, and the coloration reaction was stopped with sulphuric acid. The intensity of the coloration was proportional to the level of virus specific antibodies present in the serum. An ELISA microplate reader showed the optical density (OD) of the absorbance—an OD unit ≥ 0.2 was defined as a positive IgM. 

#### 2.4.4. Viral Isolation and Immunofluorescence Indirect Assay

Virus isolation was attempted for samples positive with RPA/RT-qPCR. A 200 µL sample was diluted at 1:10 in a Leibovitz 15 medium (L15), then added to a 25 cm^2^ flask monolayer of C6/36 cell line at 80% confluence, followed by incubation at 28 °C for 1 h, to allow virus adsorption. After incubation, the L15 medium containing 5% FBS, 1% penicillin streptomycin, and 0.05% de fungizone, was added to the flask and incubated for 10 days, or until observation of a cytopathic effect (CPE). To assess viral infection, indirect immunofluorescence (IFA) was conducted as previously described [22]. The flask content was transferred to a 15 mL tube, and clarified by low-speed (2500 rpm) centrifugation at 4 °C for 5 min. The supernatant was harvested and stored at −80 °C, while the cells were washed three times in PBS 1×, resuspended in 4 mL of PBS 1×, and then dispensed onto a glass slide. After complete drying, the cells were fixed in cold (−20°) acetone for 20 min. Staining was performed with a DENV specific hyper-immune mouse ascitic fluid, diluted at 1/40 in PBS 1×, as the first antibody. Then, the cells were incubated for 30 min with the second antibody (1/40 goat anti-mouse IgG, 1/100 Evan’s blue, diluted in PBS 1×). The slides were observed with a fluorescence microscope (Eurostar III plus, Euroimmune).

#### 2.4.5. RT-PCR Amplification, Sequencing and Phylogenetic Analysis

To define the serotype of the isolated DENV strains, viral RNA was extracted from 200 μL of DENV infected C6/36 culture supernatant, using the QIAmp viral RNA kit (Qiagen, Hilden, Germany) in accordance with the manufacturer’s instructions. The C-prM gene was amplified using the primers (DS1/DS2) described by Lanciotti et al. [23]. For cDNA synthesis, 10 μL of viral RNA was mixed with 1 μL of the random hexamer primer (2 pmol), and the mixture was heated at 95 °C for 2 min. The reverse transcription was performed in a 20 μL mixture containing 2.5 U RNasin (Promega, Madison, WI, USA), 1 μL of deoxynucleotide triphosphate (dNTP) (10 mM each DNTP), and 5 U of AMV reverse transcriptase (Promega, Madison, WI, USA), which was incubated at 42 °C for 60 min. The PCR products were generated using DS1/DS2 primers at 10 µM concentration. Five microliters of cDNA were mixed with 10× buffer, 3 μL of each primer, 5 μL of dNTPs 10 mM, 3 μL of MgCl2, and 0.5 μL of Taq polymerase (Promega, Madison, WI, USA). Amplicons of the expected size (520 bp) were purified from the agarose gel, with the QiaQuick Gel Extraction Kit (Qiagen, Hilden, Germany), as specified by the manufacturer. Both strands of each amplicon were Sanger sequenced out-of-house (Genewiz, Germany). The sequences were merged using EMBOSS Merger software, and final results were analyzed using the Basic Local Alignment Search Tool (BLAST, www.ncbi.nlm.nih.gov/) consulted on 18 July 2017. The nucleotide sequence alignments were generated using the ClustalW algorithm, and Maximum likelihood (ML) trees were inferred for each serotype using Mega software version 6 [24].

## 3. Results

### 3.1. Demographic and Clinical Data

During the study period from September 2015 to March 2016, 104 children aged between 1 and 115 months were enrolled and screened for DENV, CHIKV, ZIKV, YFV, RVFV—Seventy-nine were <5 years old, and 42 (40,38%) were male (Appendix A). Regarding clinical symptoms, the most common symptoms after fever were, respectively, rhinorrhea (65%), cough (53%; 55/104), vomiting (25%; 26/104), diarrhoea (25%; 25/104) and abdominal pain (25%; 26/104). Headaches were reported only in 10% of the enrolled patients, while myalgia and arthralgia were not recorded at all (Figure 3).

### 3.2. Molecular, Serologic and Antigenic Detection

Three samples (2.8%) tested positive with DENV1-3 RPA, while none of the other arbovirus assays yielded positive results. All of the arbovirus results were confirmed by RT-PCR, with 100% concordance. The samples of the three DENV cases were additionally tested and confirmed positive by DENV NS1 antigen capture (Table 1). The ELISA IgM test yielded negative results in all three cases (not shown). Virus isolation at 28 °C in C6/36 cells yielded non-obvious CPE ten days after inoculation. However, two strains were successfully isolated, and isolation was confirmed by IFA (Appendix A). The isolates were designated Medina Gounass 1 and Medina Gounass 2, respectively.

### 3.3. Phylogenetic Analysis

Finally, the sequence of the CprM gene of the two isolates was determined and deposited in genbank (accessions numbers MK940790 (Medina gounass 1/2015), and MK940791 (Medina gounass 1/2015)). Both strains were completely homologous (100%), with no nucleotide difference. A basic local alignment search tool for nucleotide (BLASTN), using the obtained C-prM gene sequence, showed 100% identity with the DENV-1 isolates collected in Guangzhou, China, in 2014 (KP279753.1). A calculated phylogenetic tree clustered the determined DENV-1 sequences with Asian strains, supported by high bootstrap values (Figure 4).

## 4. Discussion

The objective of this study was to assess the use of a mobile suitcase laboratory for the routine testing of arboviral etiologies of NMFI in an outpatient clinic of a suburb of Dakar, Senegal. During the six month survey of arboviral infections in febrile non malaria patients, three cases of dengue infection were detected in 104 enrolled children under 10 years old.

None of the targeted arboviruses (CHIKV, RVFV, YFV, ZIKV) except for DENV were detected during this study. The prevalence of DENV was 2.8%.

In a previous study on the etiology of acute febrile illness in Abidjan, an inter epidemic DENV prevalence of 0.4% was reported in the 812 patients tested [25]. Similarly, our work highlights an inter epidemic circulation of DENV in poor urban settings of Dakar. The difference in prevalence between the two studies may be attributable to the fact that the study conducted in Abidjan was not limited to children ≤10 years of age, as well as the smaller number of enrolled patients in our study (104 patients).

The border between interepidemic and epidemic prevalence in sub-Saharan Africa is difficult to assess, as noted by a study on febrile patients in Ibadan, Nigeria [26], which determined a prevalence of 35% of dengue infection through NS1 antigen detection. The infected patients secrete large quantities of soluble NS1 (sNS1) into the bloodstream, with concentrations of up to 50 µg/mL [27]. Soluble NS1 (sNS1) can remain in the blood for 9 days, and persist for up to 18 days in some patients [28], exceeding viremia which lasts up to 6 days. This makes NS1 a good biomarker of acute illness as it provides a wide window for DENV detection. It has been suggested that combining NS1 detection with IgM detection can outperform PCR [29]; however, the use of NS1 detection in the routine screening in dengue epidemics, as a prerequisite for hospitalization, has been questioned [30]. Additionally, fieldable ELISA systems which would allow for a comparison between the ratios of DENV NS1-Ag and DENV-RNA, are not currently available.

Phylogenetic analysis of the obtained DENV C-prM gene sequences yielded 100% identity with the isolates collected in Guangzhou, China, in 2014. A calculated phylogenetic tree clustered the determined DENV-1 sequences with Asian strains, supported by high bootstrap values (Figure 4). This finding suggests an importation of the virus to Senegal from Asia, via acutely viremic cases or by infected mosquitoes or their eggs. Indeed, in recent years, international travel and trade activity between the Asian and African continents has increased considerably [31]—between 1994 and 2009, the annual volume of trade between Senegal and China grew from 23 million U.S. dollars to 441 million U.S. dollars, representing a twenty-fold increase in 15 years [32]. The potential to extend the distribution area of individual arboviruses was recently supported by the detection of Japanese encephalitis virus (JEV), during a yellow fever outbreak in Lunda (Angola), in 2016 [33]. This virus is endemic in Asia and the western pacific, but local circulation had never been documented before in Africa [34]. Another example is the first case of RVFV infection, detected in China from a patient returning from Angola in 2017—while this virus was previously restricted to sub-Saharan Africa, it has been spreading in the Arabic peninsula since 2000 [35].

In Africa, dengue is likely to be underreported and under-recognized. This is due to the low awareness of health care providers, and the circulation of other prevalent NMFI [5]. The absence of surveillance in many African countries and the lack of effective diagnostic tools also contribute to the underestimation of the real incidence of dengue fever in Africa [31]. Since other studies report the expansion of dengue fever among NMFI [36], our study and the cited studies in Abidjan and Ibadan stress the need to implement laboratory capacity to assess the real burden of DENV in rural and urban areas of West Africa, during inter epidemic periods.

The RPA positive samples were confirmed by serological assay, viral isolation as well as real-time RT-PCR. The laboratory-based real-time RT-PCR and mobile RT-RPA results were concordant, but mobile RT-RPA yielded results in approximately 20 min, including the extraction step (Table 1). Additionally, the RPA was performed at the point of need in a suitcase laboratory. In conclusion, although virus isolation remains the “gold standard” in diagnostics [37], rapid molecular testing at the point of care can provide reliable results (short time-length process, sensitivity and specificity).

## 5. Conclusions

Our results suggest that the RT-RPA could be an alternative to real-time PCR in low resource settings. This field deployment contributed to the evaluation of the feasibility to implement point of need arbovirus diagnostics in primary care settings and showed that RT-RPA can be a reliable and accurate diagnosis tool for the detection of NMFI in low-income settings. However, studies with larger cohorts are needed.

## Figures and Tables

**Figure 1 diagnostics-10-00408-f001:**
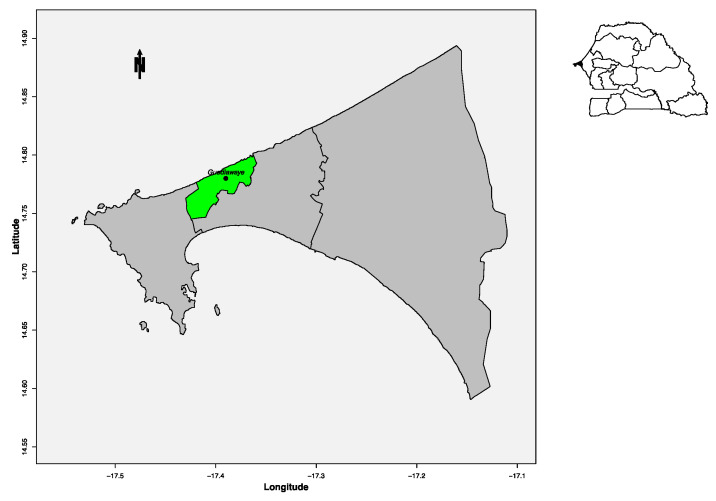
Map showing the area of study.

**Figure 2 diagnostics-10-00408-f002:**
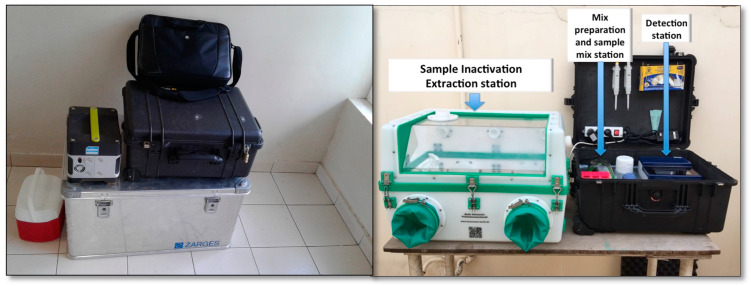
Deployment of the mobile laboratory at the Institut Pasteur de Dakar before departure (**left**), and at the site of study, Medina Gounass (**right**). For more detail on the extraction procedure, see Figure 2 in [11].

**Figure 3 diagnostics-10-00408-f003:**
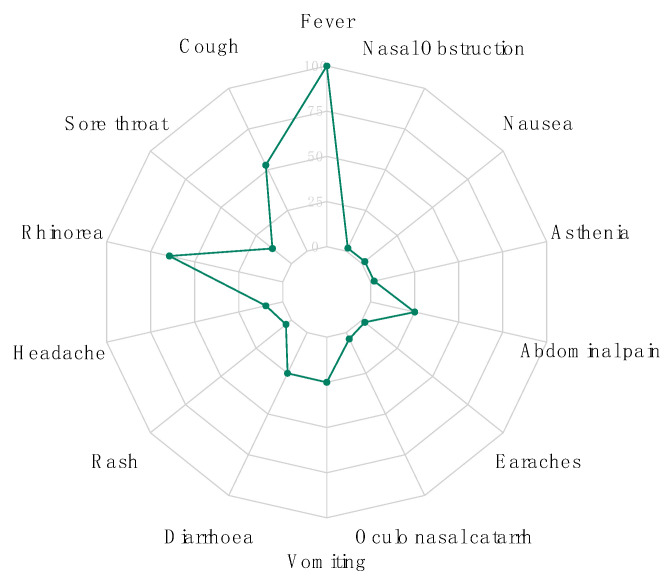
Plot showing the percentage of recorded symptoms among enrolled patients.

**Figure 4 diagnostics-10-00408-f004:**
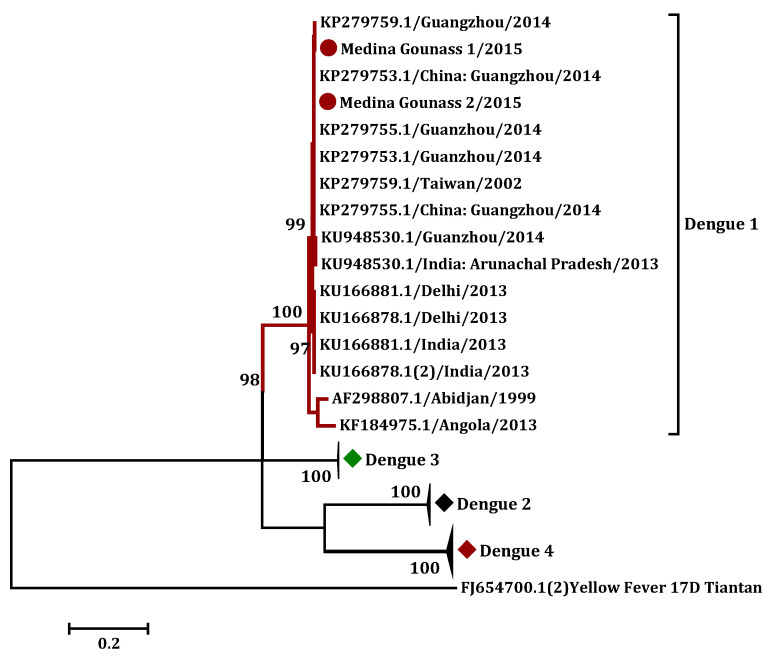
Phylogenetic tree based on the C-prM gene sequences using the maximum likelihood (ML) method, showing the relationship of 2 from 3 isolated viruses in this study (darkred circle) with 40 global samples of dengue virus (DENV), and Kimura 2 parameters; the gamma distributed (K2+G+I) nucleotide substitution model was used. The yellow fever 17D (FJ654700.1) was used as the outgroup.

**Table 1 diagnostics-10-00408-t001:** Comparison of time detection between non-structural protein 1 (NS1), real-time recombinase polymerase amplification (RT-RPA), and RT-PCR.

Sample Names	RT-qPCR		RT-RPA		NS1 Antigen Capture	
	Ct Value	Detection Time	Threshold Time	Detection Time	D.O	Average Detection Time
**Medina Gounass 1**	29.35	79 min	6 min	6 min	6.8477	2 h
**Medina Gounass 2**	27.59	75 min	5 min	5 min	7.9131	2 h
**Medina Gounass 3**	26.42	72 min	5 min	5 min	9.819	2 h

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
