# Peer review of "Mobile Laboratory Reveals the Circulation of Dengue Virus Serotype I of Asian Origin in Medina Gounass (Guediawaye), Senegal"

_diagnostics, 2020, doi:10.3390/diagnostics10060408_

Round 1

Reviewer 1 Report

In this Study, Dieng and co-authors accessed the applicability of RPA assay in the diagnosis of 5 arboviruses. The study was performed with 104 samples from children under age group 10 years. Although, the methodology used in this study shows significance in low income areas for viral diagnosis, however, there are some major flaws in the study.

Major comments:

The sample size in this study is small and limited to only children. As the arboviruses infect all age group, the study should have been conducted in different age groups and with larger sample size.

Only DENV positive samples were detected and it is not clear in the results section if the confirmation is performed with all the samples using different assays or just on the RPA positive samples. Because of the  absence of samples with any other viral infection, it is not clear how this assay will work for other viruses and its accuracy.

It is not clear what kind of probe is used, which antigen is targeted?

This study does not describe the sensitivity of the assay, what is the limit for detection?

Minor comments:

The reference list is not complete and the citation of several references is not accurate.

The data present in this study is not latest. Is it possible for authors to provide some recent data?

Do the authors have data from any other region?

Author Response

Reviewer 1

In this Study, Dieng and co-authors accessed the applicability of RPA assay in the diagnosis of 5 arboviruses. The study was performed with 104 samples from children under age group 10 years. Although, the methodology used in this study shows significance in low income areas for viral diagnosis, however, there are some major flaws in the study.

Major comments:

The sample size in this study is small and limited to only children. As the arboviruses infect all age group, the study should have been conducted in different age groups and with larger sample size.

Indeed, it is true that the sample size is small but we would like to draw the attention to the reviewer that the children were selected according to the case definition described in materials and methods. Since we tested in an interepidemic phase the detection rate was very low which actually highlights the sensitivity of the assays used.

We have updated a sentence in the introduction (line 58).

“The objective of this study was to determine potential arboviral etiologies of NMFI in children in a low resources setting using mobile recombinase polymerase amplification (RPA]) a real time isothermal amplification technique [8].”

Only DENV positive samples were detected and it is not clear in the results section if the confirmation is performed with all the samples using different assays or just on the RPA positive samples. Because of the absence of samples with any other viral infection, it is not clear how this assay will work for other viruses and its accuracy.

All samples were indeed tested with all 5 assays. We have emphasised this in line 124 which now reads:

“RNA amplification for all samples was carried out using the Tubescanner point of care device (Qiagen, Hilden, Germany) and the Twist exo RT kit (TwistDx, Cambridge, UK) using RPA primers and probes designed for the detection of 5 arboviruses “

Also see response to remark on sensitivity below.

It is not clear what kind of probe is used, which antigen is targeted?

The test used is an isothermal nucleic acid amplification test called Recombinase Polymerase Amplification which uses an exoprobe. This is clearly referenced in the introduction with reference 8. However to be more clear we have changed the relevant sentence in line 58 to:

“The objective of this study was to determine potential arboviral etiologies of NMFI in a low resources settings using mobile recombinase polymerase amplification (RPA]) a real time  isothermal amplification technique [8].”

This study does not describe the sensitivity of the assay, what is the limit for detection?

All these RPA assays were established by our group and published in peer reviewed journals. The current study is focusing on their deployment in field settings. For clarity sake we have added additional information of the analytical sensitivities of the assays to line 127

“RNA amplification was carried out using the Tubsecanner point of care device (Qiagen, Hilden, Germany) and the Twist exo RT kit (TwistDx, Cambridge, UK) using RPA amplicons designed for the detection of 5 arboviruses: Dengue virus (DENV) with two sets of primers and probes: set 1 for DENV1-3 and set 2 for DENV- 4 [10], Yellow Fever virus (YFV) [12], Chikungunya virus (CHIKV) [13], Rift Valley Fever virus (RVFV) [14], and Zika Virus (ZIKV) [15] with analytical sensitivities of 241, 14, 10, 23, 10, 21, RNA copies detected respectively.”

Minor comments:

The reference list is not complete and the citations of several references is not accurate.

                  This reference list was updated and correctly assigned

The data present in this study is not latest. Is it possible for authors to provide some recent data?

The team was very busy with haemorrhagic fever outbreaks in Guinea and DRC and a large DENV outbreak in Senegal and has not resumed screening defined cohorts for several arboviruses by RPA. However, the RPAs have served well in identifying the large DENV outbreak and in screening patients.

Do the authors have data from any other region?

                  The DENV outbreak data are being analysed separately.

Reviewer 2 Report

All comments are on the marked-up pdf

Author Response

Reviewer 2

All minor edits done

Line 26: You should mention here how many samples were collected from who?

The following sentence was added to line 35:

“One hundred and four children aged between one month and 115 months were enrolled and screened.”

Line 85:

 More photos are needed to go into details of what has been described in the protocol below.

The following sentence was added to the Figure 2 legend located at the line 116: “For more detail on the extraction procedure see Fig 2 and Fig 3 in [11].”

Line 177: I would suggest adding a section in the Results to compare the cost to other methods including sampling in the field and bringing back the samples in the lab for analysis.

A comparative analysis for the extraction costs has already been published in:

Development of Rapid Extraction Method of Mycobacterium avium Subspecies paratuberculosis DNA from Bovine Stool Samples. Hansen S, Roller M, Alslim LMA, Böhlken-Fascher S, Fechner K, Czerny CP, Abd El Wahed A. Diagnostics (Basel). 2019 Mar 29;9(2):36.

And costs have also been discussed in reference 11.

Line 186: writtings are not very clear and so are hard to read:

The graph was updated and the font size was increased. See figure line 241

Line 244: I would say infected eggs as they can hatch later and be infectious ...it is less likely that adults could survived all along until biting.

We agree with the reviewer regarding the importance of eggs transportation in virus diffusion. However, there is plenty of evidence that mosquitoes do travel in containers (Dakar has a container terminal) and that aggressive day stinging Aedes species are very efficient in spreading viruses. Therefor we have added your point side by side with the point we made.

The sentence in line 319 has been updated to:

“This finding suggests an importation of the virus to Senegal from Asia via acutely viraemic cases or by infected mosquitoes or their eggs.” In line 329

Reviewer 3 Report

The article addresses an important step forward towards improving arbovirus disease surveillance in resource poor settings. Despite the importance, the study seems to have reported quite late. It would be interesting to know whether this "mobile lab" concept has been operationalized in Africa.

The article needs sentence corrections (I have highlighted some in the attached document). Especially, many sentences have been started with numericals, which needs to be correctly spelled instead.

Please refer to the attached document for further comments.

Author Response

Reviewer 3

All minor edits done

Line 50: Since this pilot study was done 5 years, ago, has this concept been operationalized? If not, what were the limitations/challenges for not being able to implement?

The team was very busy with haemorrhagic fever outbreaks in Guinea and DRC and a large DENV outbreak in Senegal and has not resumed screening defined cohorts for several arboviruses by RPA. However, the RPAs have served well in identifying the large DENV outbreak and in screening patients. The DENV outbreak data are being analysed separately.

This concept was operationalized, and used during outbreaks including Zika outbreak in 2016 in Brazil (Abd El Wahed A, Sanabani SS, Faye O, Pessôa R, Patriota JV, Giorgi RR, Patel P, Böhlken- Fascher S, Landt O, Niedrig M, Zanotto PMA, Czerny CP, Sall AA and Weidmann M. Rapid molecular detection of Zika virus in urine using the recombinase polymerase amplification assay. Plos Curr outbreaks. 2017;9;)

Line 107: This sample volume is relatively lower than what is used in most standard extraction kits. Is there any specific reason why the samples volume was kept so low? This could contribute to low RNA yield that affects downstream detection.

Yes this is a modified assay validated and used routinely in the lab in Dakar. The reason is that most of the time we received low sera volumes during arbovirus monitoring and we optmised the extraction in order to retain enough for bio-banking and others experiments.

An additional sentence was added to the methods section from line 144:

“Viral RNA was extracted from 100 μL of human sera using the QIAmp viral RNA kit (Qiagen, Hilden, Germany) according to the manufacturer’s instructions. The reduced input of 100µl serum for this kit has been validated in house.”

Line 86: This sample volume is relatively lower than what is used in most standard extraction kits. Is there any specific reason why the samples volume was kept so low? This could contribute to low RNA yield that affects downstream detection.

This is the recommended volume by the manufacturer. We have used this very efficiently in swab extraction for EVD cases in Guinea. Please refer to reference 9.

Line 87: Which type of beads were used?

The beads were provided with the kit SpeedXtract Nucleic Acid Kit (SE), Qiagen, Hilden, Germany) as specify in the line 113. These beads allow a reverse extraction method by capturing cell membrane, protein and other inhibitors in the sample, while the nucleic acid remains in the supernatant.

Line 237

I am not sure what authors want to highlight here. NS1 is widely used as a rapid diagnostic tool in many dengue endemic settings. It is an useful tool to detect acute infections.

The correspondence cited describes that only 54% of acute cases were NS1 Ag ELISA positive and most of them were non-severe cases. They conclude that the role of rapid diagnostic NS1 antigen tests during a dengue fever epidemic should not be a criteria for hospital admissions.

To be more precise we have updated the sentence in line 321:

“…however, the use for NS1 detection in routine screening in dengue epidemics as a prerequisite for hospitalization has been questioned (28). “

Line 241: Based on the tree, it seems to me that study sequences are identical to those from China (2014).

We updated the sentence in line 326 to:

Phylogenetic analysis of the obtained DENV C-prM gene sequences yielded 100% identity with isolates collected in Guangzhou, China in 2014”

Line 320: It is better to discuss more about the trade and travel connections between Senegal (esp. study area) and China because DENV-1 viruses show close relatedness to those from Guangzhou.

Documents assessing weight of trade and travel frequencies between china and Senegal are scarce.

But we added the following sentences in line 331:

“Indeed between 1994 and 2009 the annual volume of trade between Senegal and China grew from 23 million U.S. dollars to 441 million U.S. dollars representing a twentyfold increase in 15 years (30).”

Line 263: This statement totally underestimates the purpose of this study. Though virus isolation is still the gold standard to confirm infections, the method is practically challenging in resource poor settings. Moreover, virus isolation is not rapid. This is the reason why other reliable alternative methods are becoming more popular and appropriate for routine diagnostics.

The paragraph has been changed to in line 352

“Additionally, the RPA was performed at the point of need in a suitcase laboratory. In conclusion, although virus isolation remains the "gold standard" in diagnostics [37] rapid molecular testing at the point of care can provide reliable results (short time-length process, sensitivity and specificity).”

Line 266: The important conclusion in this sentence is ease of setting up RT-RPA detection in resource poor settings. Given the low positivity rate, the data does not provide substantial evidence to state whether RT-RPA is reliable and provides accurate diagnosis.

An additional sentence was added to the paragraph in line 416:

“Our results suggest that the RT-RPA could be an alternative to real-time PCR in low resource settings. This field deployment contributed to the evaluation of the feasibility to implement point of need arbovirus diagnostics in primary care settings and showed that RT-RPA can be a reliable and accurate diagnosis tool for the detection of NMFI in low-income settings. However, studies with larger cohorts are needed.”

Round 2

Reviewer 1 Report

Authors have replied to all the comments on the manuscript. I believe, this manuscript can be accepted for publication in Diagnostics journal.